# Mitochondrial Kv1.3 Channels as Target for Treatment of Multiple Myeloma

**DOI:** 10.3390/cancers14081955

**Published:** 2022-04-13

**Authors:** Stephanie Kadow, Fabian Schumacher, Melanie Kramer, Gabriele Hessler, René Scholtysik, Sara Oubari, Patricia Johansson, Andreas Hüttmann, Hans Christian Reinhardt, Burkhard Kleuser, Mario Zoratti, Andrea Mattarei, Ildiko Szabò, Erich Gulbins, Alexander Carpinteiro

**Affiliations:** 1Institute of Molecular Biology, University Hospital Essen, University of Duisburg-Essen, 45147 Essen, Germany; stephanie.kadow@uk-essen.de (S.K.); melanie.kramer@uk-essen.de (M.K.); gabriele.hessler@uni-due.de (G.H.); erich.gulbins@uni-due.de (E.G.); 2Institute of Pharmacy, Freie University Berlin, 14195 Berlin, Germany; fabian.schumacher@fu-berlin.de (F.S.); kleuser@zedat.fu-berlin.de (B.K.); 3Institute of Cell Biology, University Hospital Essen, University of Duisburg-Essen, 45147 Essen, Germany; rene.scholtysik@uk-essen.de; 4Department of Hematology and Stem Cell Transplantation, West German Cancer Center, University Hospital Essen, 45147 Essen, Germany; sara.oubari@uk-essen.de (S.O.); patricia.johansson@uk-essen.de (P.J.); andreas.huettmann@uk-essen.de (A.H.); christian.reinhardt@uk-essen.de (H.C.R.); 5Department of Biomedical Sciences, University of Padova, 35121 Padova, Italy; zoratti@bio.unipd.it; 6CNR Institute of Neuroscience, 35121 Padova, Italy; ildiko.szabo@unipd.it; 7Department of Pharmaceutical and Pharmacological Sciences, University of Padova, 35121 Padova, Italy; andrea.mattarei@unipd.it; 8Department of Biology, University of Padova, 35121 Padova, Italy

**Keywords:** Kv1.3, multiple myeloma, ABT-199, venetoclax, mitochondria

## Abstract

**Simple Summary:**

Multiple myeloma is a non-curable disease and new therapeutic approaches are needed. PAPTP and PCARBTP, two novel mitochondria-specific inhibitors of the Kv1.3 ion channel, are effective in killing cultured myeloma cell lines and myeloma cells isolated from patient punctates, while healthy bone marrow cells are not affected. Cell death occurs through the classical mitochondrial apoptotic pathway, and further treatment with venetoclax, a BCL-2 inhibitor, has a clear synergistic effect. We identify Kv1.3 channels as a new therapeutic target for the treatment of multiple myeloma.

**Abstract:**

Despite several new developments in the treatment of multiple myeloma, all available therapies are only palliative without curative potential and all patients ultimately relapse. Thus, novel therapeutic options are urgently required to prolong survival of or to even cure myeloma. Here, we show that multiple myeloma cells express the potassium channel Kv1.3 in their mitochondria. The mitochondrial Kv1.3 inhibitors PAPTP and PCARBTP are efficient against two tested human multiple myeloma cell lines (L-363 and RPMI-8226) and against ex vivo cultured, patient-derived myeloma cells, while healthy bone marrow cells are spared from toxicity. Cell death after treatment with PAPTP and PCARBTP occurs via the mitochondrial apoptotic pathway. In addition, we identify up-regulation of the multidrug resistance pump MDR-1 as the main potential resistance mechanism. Combination with ABT-199 (venetoclax), an inhibitor of Bcl2, has a synergistic effect, suggesting that mitochondrial Kv1.3 inhibitors could potentially be used as combination partner to venetoclax, even in the treatment of t(11;14) negative multiple myeloma, which represent the major part of cases and are rather resistant to venetoclax alone. We thus identify mitochondrial Kv1.3 channels as druggable targets against multiple myeloma.

## 1. Introduction

In Europe, the incidence of multiple myeloma is 7.7/100,000 in males and 6.0/100,000 in females (http://gco.iarc.fr/today accessed 3 April 2022). Patients are usually between 55 and 90 years of age (http://gekid.de/atlas accessed 3 April 2022). Multiple myeloma is a malignancy of plasma cells, which are primarily localized in the bone marrow [1].

Multiple myeloma is a heterogeneous disease, which develops from the precursor condition called monoclonal gammopathy of undetermined significance (MGUS) as a result of the accumulation of oncogenic events. These mutations may include, among others, primary IgH translocations, hyperdiploidy, dysregulation of cyclin D1, MYC deregulation, chromosome 13 deletions, loss of chromosome 17p, activating mutations of RAS or B-RAF and further secondary translocations [2]. According to the International Myeloma Working Group the occurrence of t(4;14), t(14;20), t(14;16), del 17p, non-hyperdiploidy and gain(1q) in multiple myeloma cells predicts high-risk for early progression and shortened overall survival after therapy defining the group of patients with high-risk cytogenetics [3]. Patients with high-risk cytogenetics show lower response rates to therapy indicating that also the sensitivity of myeloma cells themselves to anti-myeloma agents may be reduced in the case of high-risk cytogenetics [4]. Until the early 2000s, chemotherapies with mainly alkylators and anthracyclines and corticosteroids were the basis of myeloma therapy. Since 2005 new groups of drugs as IMiDs, proteasome-inhibitors, anti CD38-antibodies and antibody–drug conjugates as mono-agents or in combinations are used in clinical routine. Further new drugs and CAR-T cell strategies are being tested in clinical trials—recently first CAR-T cell therapies were admitted in the relapsed/refractory settings. In patients with t(11;14) venetoclax, a specific Bcl-2 inhibitor, has shown efficacy as targeted therapy in the relapsed/refractory setting [5]. In patients younger than 65–70 years, high-dose chemotherapy with autologous stem cell transplantation is of clinical standard. Better therapeutic options and improvements of supportive care ameliorated outcomes and prognosis, in particular in the group of younger patients (<65 years), but the median survival is still only 7–8 years. However, in patients older than 70 years the median survival is less than 2 years [6]. Since all of the available therapies are only palliative without curative potential all patients ultimately relapse. Thus, novel therapeutic options are urgently required to prolong survival of or to even cure myeloma.

Mitochondria play essential roles in numerous cellular functions and in homeostasis, such as ATP production, the regulation and initiation of apoptosis, and in several metabolic adaptations. Multiple myeloma cells generally have upregulated protein biosynthesis as part of immunoglobulin synthesis and myeloma cells are therefore dependent on degrading misfolded, dysfunctional synthesized proteins as part of the Unfolded Protein Response [7,8]. Treatment with proteasome inhibitors, such as bortezomib or carfilzomib, results in decreased degradation of these misfolded, dysfunctional proteins, leading to proteotoxic stress, and subsequently to activation of the Unfolded Protein Response, leading to apoptosis [7]. It has been shown in vitro that cell lines resistant to proteasome inhibitors have a distinctly upregulated chaperone machinery to process the misfolded proteins, for which adaptations in energy and mitochondrial metabolism are also necessary and have been described [9]. In addition to these changes, upregulation of the multi-drug resistance protein (MDR-1) ABCB1 was detected, particularly in cell lines where resistance to carfilzomib was generated, and carfilzomib resistance could be abrogated by inhibition of MDR-1 [9].

Kv1.3 is expressed in many organs and cells under physiological conditions, for instance in immune cells, but the channel is also expressed in many cancer cells, including melanoma, leukemia, myeloma, pancreatic and breast cancer [10,11,12,13]. In fact, all tumors we tested so far expressed Kv1.3, except osteosarcoma cells. The channel is expressed in the plasma membrane and in mitochondria, as well in cis-Golgi and the nuclear membrane. While plasma membrane Kv1.3 in general seems to promote proliferation, mitochondrial Kv1.3 participates in cell death [10,11,12,13,14,15,16,17,18].

We demonstrated a dual localization of Kv1.3 in the plasma membrane and the inner mitochondrial membrane of lymphocytes [14]. This finding was confirmed by several groups and was also shown for other channel proteins [15,16]. We also demonstrated that direct inhibition of mitochondrial Kv1.3 induces cell death. We showed that Bax and Bak physically interact with mitochondrial Kv1.3 via lysine 128 that folds into the pore of Kv1.3 and thereby inhibits the channel. Kv1.3 inhibition results in transient mitochondrial hyperpolarization, release of mitochondrial reactive oxygen species, release of cytochrome c, opening of the permeability transition pore and finally apoptosis [17,18]. Genetic deficiency or down-regulation of Kv1.3 prevented death induced by a variety of stimuli, including Bax and Bak [14,17,18]. We also applied these concepts to malignant tumors. Employing membrane-permeable Kv1.3 inhibitors such as clofazimine, Psora 4 and PAP-1 [19,20], we showed that these Kv1.3 inhibitors efficiently kill chronic lymphocytic leukemia (B-CLL) cells. B cells from healthy individuals and residual normal T lymphocytes from B-CLL patients were spared and not killed [20,21].

Since Psora-4, PAP-1, and clofazimine do not specifically target mitochondrial Kv1.3, we synthesized novel inhibitors of Kv1.3 that are specifically targeted to mitochondria. These mitochondriotropic derivatives of PAP-1 contain a lipophilic, positively charged triphenylphosphonium group (TPP+) mediating accumulation of the compound in regions with negative electrical potential, in particular mitochondrial (Δψm ranges between −150 and −180 mV) [13]. In one compound, PAPTP, the TPP+-containing chain is linked to the rest of the molecule by a chemically stable C-C bond. In the other drug, PCARBTP, TPP+ is linked to the PAP-1 core via a carbamic ester bond O-C(O)-N, which is hydrolyzed in the cell to release PAPOH. This product represents a prodrug and differs from PAP-1 only by the presence of a hydroxyl group. We proved that the TPP+ moiety efficiently drives rapid uptake of the drugs into isolated mitochondria [13]. The inhibitors induced death exclusively in cells expressing Kv1.3 but did not affect survival of cells lacking Kv1.3. In contrast to the membrane-impermeant Kv1.3 inhibitor margatoxin, PAPTP and PCARBTP reduced cell viability and efficiently killed B16F10 melanoma cells, underlining the notion that the novel Kv1.3 inhibitors act by targeting Kv1.3 in mitochondria. Further, PCARBTP and PAPTP preferentially killed malignant cells, but did not affect healthy ex vivo primary human B cells or other leukocytes, since the ROS-inducing effect of these drugs is additive to a high basal ROS level, generally observable in cancer cells [13]. In orthotopic mouse models of two poor-prognosis cancers, melanoma and pancreatic ductal adenocarcinoma, the compounds reduced tumor size by more than 90% and 60%, respectively, while sparing immune and cardiac functions [13].

Since multiple myeloma cells also express high levels of Kv1.3 [22] (also see https://www.proteinatlas.org/ENSG00000177272-KCNA3/single+cell+type (accessed on 25 February 2022) for RPMI-8226) and are characterized by high intracellular ROS levels [23], here we investigated the effects of these novel Kv1.3 inhibitors on different multiple myeloma cell lines as well as in patient-derived primary multiple myeloma cells.

## 2. Materials and Methods

### 2.1. Human Cell Lines and Generation of Resistant Cells

Human multiple myeloma cell lines, RPMI-8226 and L-363, were purchased from the German Collection of Microorganisms and Cell Cultures GmbH (DSMZ, Braunschweig, Germany). All cells were cultured in RPMI-1640 medium (Gibco; Thermo Fisher Scientific Inc., Waltham, MA, USA) supplemented with 10% (RPMI-8226) or 15% fetal bovine serum (L-363) (Gibco; Thermo Fisher Scientific Inc., Waltham, MA, USA) and 1% penicillin and streptomycin at 37 °C in a humidified incubator containing 5% CO_2_.

Resistant RPMI-8226 and L-363 cells were generated by culture with increasing amounts of either PAPTP or PCARBTP according to an established protocol [24]. The starting dosage of PAPTP or PCARBTP was 1/10 of the EC50, medium was exchanged every other day. Every 1–2 weeks, concentrations of PAPTP or PCARBTP were increased by 20%. Finally, cells were designated as resistant to a certain dosage when (i) the resultant cell line grew exponentially in the continuous presence of the respective concentrations of PCARBTP (1 µM in the case of RPMI-8226R1.0 and 3.5 µM in the case of L-363R3.5) and (ii) showed a long-term stability after removal of PCARBTP for several weeks (up to 90 days) and (iii) remained resistant after freeze/thaw cycles [25].

### 2.2. Cell Death Detection and Multiple Staining

After treatment with the various compounds, apoptotic cells were identified by labeling with Annexin V-APC/7-AAD (both Biolegend Inc., San Diego, CA, USA) for 15 min at RT. Cells were then collected and analyzed by flow cytometry (Attune NxT flow cytometer, Thermo Fisher Inc., Waltham, MA, USA). Dead cells were calculated by determining Annexin V^+^/7AAD^+^ cells in treated samples relative to a solvent control or untreated control (bone marrow samples). In multiple-staining experiments, cells were incubated on ice with CD19-FITC (Beckman Coulter, Brea, CA, USA, #A07768), CD38-PE (Beckman Coulter, Brea, CA, USA, #A07779), CD138-PE/Cy7 (Biolegend Inc., San Diego, CA, USA, #352318), CD45-AF700 (Beckman Coulter, Brea, CA, USA, #A79390), CD56-APC/Fire750 (Biolegend Inc., San Diego, CA, USA, #362554) and CD269-BV421 (Biolegend Inc., San Diego, CA, USA, #357520) for 20 min prior to Annexin V-APC/7-AAD staining. Multiple myeloma cells were identified as CD38^+^, CD138^+^, CD19^−^, CD56^+^, CD269^+^, CD45^+/−^.

### 2.3. Mitochondria Enrichment and Western Blot

Mitochondria enrichment was performed using differential centrifugation. Cell pellets were resuspended in ice-cold homogenization medium CHM (150 nM MgCl_2_, 10 mM KCl, 10 mM Tris-Cl, pH 6.7) and left on ice for 2 min. Cells were homogenized with a pestle in a Dounce homogenizer; thereafter, 1/3 vol ice-cold CHM containing sucrose (final conc. 0.25 M) was added and mixed gently. Nuclei were pelleted by centrifugation for 5 min at 1000× *g*, 4 °C. Supernatant was centrifuged for 10 min at 5000× *g*, 4 °C and resulting pellet was resuspended ice-cold sucrose/Mg^2+^ medium using two or three gentle strokes of the pestle in a Dounce homogenizer and recentrifuged at 5000× *g*, 10 min. The resulting pellet was resuspended in RIPA-Buffer (50 mM Tris-HCl (pH7.4), 150 mM NaCl, 1 mM PMSF, 1 mM EDTA, 5 µg/mL aprotinin, 5 µg/mL leupeptin, 1% Triton x-100, 1% sodium deoxycholate, 0.1% SDS). Protein concentration was determined using Bradford reagent (Bio-Rad Laboratories Inc., Hercules, CA, USA). For Kv1.3 detection in whole cell lysates 50 µg protein, for enriched mitochondria samples, 30 µg protein was applied per lane. Proteins were loaded and separated in an 8.5% SDS-PAGE gel and transferred onto a nitrocellulose membrane (GE Healthcare, Chicago, IL, USA). Membranes were blocked with 4% BSA/PBS for 1 h and incubated with anti-Kv1.3-antibody (Alomone Labs, Jerusalem, Israel, #APC-101) and anti-Tim23-antibody (BD, Franklin Lakes, NJ, USA, #611223). Horseradish peroxidase or alkaline phosphatase conjugated secondary antibodies (both Cell signaling, Danvers, MA, USA) were detected by chemiluminescence (Pierce, Thermo Fisher Scientific Inc., Waltham, MA, USA) using film (Amersham, GE Healthcare Limited, Pollards Wood, Bucks, UK).

### 2.4. Mitochondrial Membrane Potential and ROS Production

Mitochondrial membrane potential was monitored using tetramethylrhodamine methyl ester (TMRM); ROS production was followed using MitoSOX Red (both Thermo Fisher Inc., Waltham, MA, USA). Cells were incubated for 30 min at 37 °C with either 20 nM TMRM or 2.5 μM MitoSOX and were washed twice. After incubation, the desired compounds were added (as indicated in the figure legends), and the decrease in TMRM fluorescence or the increase in MitoSOX fluorescence was measured by FACS at various time points over 2 h.

### 2.5. Caspase Activation

Caspase 3/7 activation was measured by using CellEvent™ Caspase 3/7 Green Ready Flow™ Reagent (Invitrogen, Thermo Fisher Inc., Waltham, MA, USA) according to the manufacturer`s instructions. Briefly, cells were treated with PAPTP, PCARBTP, staurosporine or solvent control as indicated. A total of 30 min prior to the indicated treatment-time, CellEvent™ Caspase-3/7 green Ready-Flow™-reagent was added to the cells and analyzed by flow cytometry (Attune NxT flow cytometer, Thermo Fisher Inc., Waltham, MA, USA).

### 2.6. RNA Sample Preparation and Sequencing

Total RNA was extracted from parental and resistant RPMI-8226 and L-363-cells in triplicate using the QIAGEN RNeasy mini kit and included the optional digestion with RNAse-free DNAse I (QIAGEN, Hilden, Germany). RNA quality was analyzed on Agilent Bioanalyzer RNA nano chips (Agilent Technologies, Inc., Santa Clara, CA, USA) and revealed RIN values between 9.5 and 9.9. An amount of 50 ng of total RNA was used to generate NGS libraries using the QuantSeq 3′ mRNA FWD kit (Lexogen, Vienna, Austria), which generates libraries of sequences close to the 3′ end of poly(A) RNAs. Libraries with unique barcodes were pooled, quantitated by qPCR and subjected to single-read sequencing on a NextSeq 500 MidOutput flowcell (Illumina, San Diego, CA, USA) as recommended by the manufacturer. After demultiplexing, reads were subjected to the automated QuantSeq data analysis pipeline provided by Lexogen. In a first step, the pipeline performed trimming, quality control, alignment, and read counting from fastqc input. Thereafter, differentially expressed genes were identified by the differential expression (DE) QunatSeq pipeline (Lexogen, Vienna, Austria).

### 2.7. MDR-1 Inhibition

Parental and resistant cells (RPMI-8226 and L-363) were cultured with PAPTP, PCARBTP or solvent control and 100nM elacridar (specific for MDR-1/BCR inhibition), (Sigma-Aldrich, St. Louis, MO, USA) as indicated in the figure legends. Cell death was measured 24 h later by Annexin V-APC and 7-AAD staining and analyzed by flow cytometry (Attune NxT flow cytometer, Thermo Fisher Inc., Waltham, MA, USA).

### 2.8. Synergy Calculation

RPMI-8226 and L-363 cells were cultivated (as indicated in the figure legends) for 24 h. Cells were harvested and stained with Annexin V-APC and 7-AAD (Biolegend Inc., San Diego, CA, USA) for cell death detection. Synergies were calculated using the online tool https://synergyfinder.fimm.fi accessed on 24 February 2022 [26] and expressed as the zero-interaction potency-Score [27].

### 2.9. Patient-Derived Bone Marrow Samples

Patient-derived multiple myeloma cells were collected from fresh bone marrow samples which were leftovers from diagnostic punctures performed in the context of standard care, after obtaining informed consent from the patients and approval by the institutional ethical committee (reference number 19-8786-BO, Ethics Committee of the Medical Faculty of the University of Duisburg-Essen). Heparinized bone marrow samples were mixed with PBS 1:1 and mononuclear cells were isolated by Histopaque^®^ 1077 (Sigma-Aldrich, St. Louis, MO, USA) density gradient centrifugation. Mononuclear cells were cultured in IMDM (Iscove’s Modified Dulbecco’s Medium, Gibco, Thermo Fisher Inc., Waltham, MA, USA) with 20% FCS at a concentration of 2 × 10^6^/mL and incubated with PAPTP, PCARBTP as indicated or 0.1% DMSO (solvent control). After 24 h, cells were harvested and stained with Annexin V-APC and 7-AAD (Biolegend Inc., San Diego, CA, USA) for cell death according to the manufacturer’s instructions and simultaneously for CD19, CD38, CD45 (Beckman Coulter, Brea, CA, USA) and CD56, CD138, CD269 (Biolegend Inc., San Diego, CA, USA) to distinguish regular bone marrow/blood cells from malignant plasma cells.

Samples from patients were collected between June 2019 and April 2021. Twenty patients were initial diagnoses of multiple myeloma and 7 patients had relapsed or progressed disease. The relapsed or progressive patients were previously treated with 1 to 4 lines of therapy, the age of the patients ranged from 37 to 79 years, with a median age of 62 years. A total of 6 of the 7 pretreated patients had been previously induced with bortezomib-containing therapy and subsequently received high-dose therapy with autologous hematopoietic stem cell transplantation and maintenance therapy with lenalidomide as part of primary treatment. A total of 3 of the 7 patients received combination therapies with daratumumab and IMiDs or proteasome inhibitors in the primary or relapse therapy.

### 2.10. Statistics

All statistic calculations were performed using GraphPad Prism version 9.3 for Windows (GraphPad Software, San Diego, CA, USA). EC50 values were calculated choosing Nonlinear Regression, Dose–response-Inhibition, log (Inhibitor) vs. normalized response—variable slope. Flow cytometry data were further analyzed by FlowJo Software version 10.7.2 for Windows (BD, Franklin Lakes, NJ, USA).

## 3. Results

First, we measured Kv1.3 expression in mitochondria in human myeloma cell lines RPMI-8226 and L-363 and, as positive control, in Jurkat cells. We found pronounced but variable expression of Kv1.3 in both human myeloma cell lines (Figure 1a).

To test the effects of mitochondrial Kv1.3 inhibitors in multiple myeloma cells, we incubated the human multiple myeloma cell lines RPMI-8226 and L-363 with increasing concentrations of PAPTP and PCARBTP (range 0.05 to 10 µM) for 24 h (Figure 1b). As control, cells were incubated with the solvent, i.e., 0.1% DMSO. As a second control to determine the effect of Kv1.3 inhibition expressed in plasma cell membrane, we incubated the cells with the cell membrane-impermeable Kv1.3 blocker margatoxin (2 µM). As a positive control for apoptosis, we incubated the cells with staurosporine (2 µM), a very strong inducer of intrinsic apoptosis. We determined cell death by staining with Annexin V-APC and 7-AAD and flow cytometry analysis. Inhibition of Kv1.3 channels located in the plasma cell membrane with margatoxin, a cell membrane impermeant inhibitor of Kv1.3, or treatment with the solvent DMSO did not induce cell death. Single treatment with the mitochondria-targeted Kv1.3 inhibitors PCARBTP or PAPTP, respectively, efficiently killed human multiple myeloma cells lines with EC50 ranging between 0.1 and 0.9 µM depending on cell line and inhibitor (Figure 1b). In detail, PAPTP and PCARBTP efficiently killed the multiple myeloma cell lines RPMI-8226 (EC50 0.1 µM PAPTP/0.26 µM for PCARBTP) and L-363 (0.29 µM/0.87 µM), respectively.

To test the effects of mitochondria-targeted Kv1.3 inhibitors on patient-derived multiple myeloma cells we collected fresh bone marrow samples from patients with multiple myeloma. We analyzed survival of multiple myeloma cells after 24 h incubation with increasing doses of PCARBTP and PAPTP. Samples were analyzed by flow cytometry using 7-AAD and Annexin V-APC staining of the cells. Tumor cells were identified as CD38^+^, CD138^+^, CD19^−^, CD56^+^, CD269^+^, CD45^+/−^. Since the samples also contained healthy, regular hematopoietic and blood cells, we determined the effect of any treatment on normal, non-malignant cells in the same bone-marrow samples (Figure 2). We analyzed samples from first diagnosis patients and patients with relapsed/refractory myeloma. Further, we differentiated between the presence of high risk or standard risk cytogenetics in samples from first diagnosis patients [3].

The results show that more than 70–90% of malignant plasma cells died by one single treatment with 1 µM PAPTP or 2.5 µM PCARBTP, while non-malignant bone marrow cells (progenitor cells of different differentiation, B-cells, T-cells, macrophages, etc.) were almost completely resistant. We further tested dosages of PAPTP and PCARBTP up to 10 µM and did not detect relevant toxicity on non-malignant cells (not shown). The effects on tumor cells were independent from the presence of high-risk cytogenetics in untreated patients or in the condition of relapsed/refractory multiple myeloma (Figure 2). The plasma cells from 13 out of 14 tested patients with first diagnosis of multiple myeloma and standard-risk cytogenetics were sensitive to PAPTP and PCARBTP. The EC50 of PAPTP and PCARBTP ranged between <0.001 µM and 2.2 µM for PAPTP and <0.001 µM and 5.53 µM for PCARBTP. All patients with first diagnosis of multiple myeloma and high-risk cytogenetics were sensitive to PAPTP, the EC50 ranged between 0.02 µM and 4.8 µM, whereas 5 out of 6 patients were sensitive to PCARBTP, here the EC50 ranged between 0.06 and 4.43 µM. In relapsed patients, all patients were sensitive to PAPTP and PCARBTP. The EC50 of PAPTP and PCARBTP ranged between 0.02 µM and 3.36 µM for PAPTP and 0.002 µM and 7.82 µM for PCARBTP (Appendix A). In this study 20 patients with first diagnosis of multiple myeloma were included. For one patient we did not have sufficient data because he received further treatment elsewhere and another patient only received local radiotherapy; therefore, therapy response after chemotherapy and survival could only be assessed for the remaining 18 patients, we were able to review the hematologic response to induction therapy and the best hematologic responses. The median follow-up time was 17 months, and at the time point of analysis 17/18 patients were alive. All patients achieved at least a very good partial remission. We found no correlation between treatment response and in vitro sensitivity to Kv1.3 inhibition.

To define the signaling events triggered by PAPTP and PCARBTP leading to cell death, we measured drug-triggered changes in mitochondrial membrane potential, release of reactive oxygen species (ROS) and activation of caspases 3/7.

We detected an early increase in mitochondrial ROS and breakdown of the membrane potential in mitochondria in RPMI-8226 and L-363 cells (Figure 3a,b). Further, we detected the activation of caspase 3/7 as hallmarks of apoptotic cell death (Figure 3c).

Resistance to known treatments of malignant cells is a general problem in cancer therapy, often leading to disease relapse. To identify molecular mechanisms that may mediate resistance to mitochondrial Kv1.3 inhibitors PAPTP and PCARBT in multiple myeloma and to counteract them, we aimed to generate PCARBTP- and PAPTP-resistant cell lines. Therefore, RPMI-8226 and L-363 cells were cultured with increasing amounts of PAPTP or PCARBTP. For PCARBTP, we were able to establish RPMI cells that were resistant up to 1.0 µM (RPMI-8226-R1.0) and L-363 cells that were resistant to PCARBTP up to 3.5 µM (L-363-R3.5) (Figure 4a,b). The EC50 of PCARBTP in RPMI-8226-R1.0 cells increased to 2.36 µM in L-363-R3.5 cells to >4 µM. However, culture of L-363 and RPMI-8226 cells with PAPTP as above described did not result in resistance against PAPTP despite several attempts.

To examine resistance mechanisms to PCARBTP in our generated cell lines, we performed RNA-sequencing analysis for resistant L-363-R3.5 and RPMI-8226-R1.0 cells and the respective parental cells (Appendix A). For L-363/L-363-R3.5 19,888 genes were analyzed, of which 7 genes were significantly (*p* < 0.01) up-regulated and 43 were down-regulated in the resistant cell line. For RPMI-8226/RPMI-8226-R1.0 22,868 genes were analyzed, of which 74 genes were significantly up-regulated and 183 were down-regulated in the resistant cell line. Among all the up- or down-regulated genes (+/− 2 fold log2 change, *p* < 0.01), ABCB1 (MDR-1), a gene encoding p-glycoprotein, also called MDR-1 protein, was strongly up-regulated in both resistant cell lines: we found a 5.12 log2-fold change in L-363-R3.5 cells and a 4.0 log2-fold change in RPMI-8226-R1.0 cells compared with parental L-363 and RPMI-8226 cells. MDR-1 is a membrane efflux pump and responsible for decreased drug accumulation in multidrug-resistant cells and often mediates the development of resistance to anticancer drugs. Since PCARBTP is a substrate for MDR-1 [13], upregulation of the gene seemed a very likely resistance mechanism to PCARBTP in our resistant cell lines.

To test whether MDR-1 protein is responsible for the resistance to PCARBTP in resistant cell lines we inhibited MDR-1 by incubation of the cells with 100 nM elacridar, a specific inhibitor of MDR-1. In both resistant cell-lines, co-administration of elacridar with PCARBTP nearly completely restored the sensitivity to PCARBTP without further sensitizing the parental cells (Figure 4a,b). Our data show that resistance that occurs under treatment with PCARBTP is mostly driven by the upregulation of MDR-1, since inhibition of the MDR-1 protein almost completely restores sensitivity to PCARBTP in resistant cells.

We next investigated whether the PCARBTP-resistant cell lines RPMI-8226-R1.0 and L-363-R3.5 showed cross-resistance to the effect of PAPTP. We incubated the PCARBTP-resistant cell lines with increasing doses of PAPTP in the presence or absence of elacridar and determined the EC50 values. The RPMI-8226-R1.0 cells showed a slightly increased EC50 (0.19 µM vs. 0.09 µM) compared to their parental cells, and the cells could be completely re-sensitized against PAPTP by additional administration of elacridar (Figure 4c). The L-363-R3.5 cells also showed a slightly increased EC50 (0.75 µM vs. 0.29 µM) compared to their parental cells. By additionally administering elacridar, the cells could again be completely re-sensitized against PAPTP (Figure 4d). These data show that PCARBTP-resistant cells also show a certain degree of cross-resistance to PAPTP, and that sensitivity can be completely restored by inhibition of MDR-1 by elacridar.

In an additional approach to further improve the effect of PAPTP, we combined PAPTP with ABT-199, a blocker of Bcl-2-proteins [28], since previous studies from our group have shown that Bcl-2-like proteins interact with Kv1.3 [17,18] and therefore might influence the effect of PAPTP on Kv1.3. In addition, in multiple myeloma patients, a correlation is found between expression of Kv1.3 and anti-apoptotic Bcl-2 (Multiple Myeloma HOVON-87/NMSG-18-trail [29] dataset obtained from 180 patients, publicly available). We therefore tested whether there is synergy between PAPTP and ABT-199 in inducing apoptosis in L-363 and RPMI-8226 cells. We treated cells with different concentrations of PAPTP and ABT-199 and determined the percentage of apoptotic, Annexin V^+^/7-AAD^+^ cells (Figure 5a) for each combination. Synergy was expressed as the zero-interaction potency (ZIP)-Score [27] between the two drugs. Briefly, the synergy-score expresses the proportion in percent of apoptotic cells that can be attributed to the drug interactions and not merely by the additive effect of PAPTP and ABT-199. A score larger than 10 is considered as synergy. In the case of RPMI-8226 and L-363 cells we find, depending on the applied concentrations, very high synergy scores up to 50 to 60 (Figure 5b).

## 4. Discussion

We identify mitochondrial Kv1.3 channels as new potential targets for the treatment of multiple myeloma. We have shown that Kv1.3 channels are expressed in mitochondria of the multiple myeloma cell lines L-363 and RPMI-8226 (Figure 1a). Treatment of these cell lines with the mitochondria-targeted Kv1.3 inhibitors PAPTP and PCARBTP leads to cell death. The EC50 ranges between 0.09 µM and 0.29 µM for PAPTP and between 0.26 µM and 0.87 µM for PCARBTP (Figure 1b,c). Further, we have shown that the efficacy of PAPTP and PCARBTP in inducing cell death also applies to ex vivo-cultured plasma cells from multiple myeloma patients (Figure 2). We determined the EC50 of malignant, patient-derived plasma cells treated with PAPTP and PCARBTP (Appendix A). Comparison of EC50 values for plasma cells from patients with first diagnosis/standard risk with first diagnosis/high-risk [3] or relapsed/refractory multiple myeloma revealed no significant differences in statistical tests (not shown). Increasing the dose of PAPTP and PCARBTP did not result in measurable toxicity in healthy bone marrow cells. This result is in line with the observations of previously published data in blood cells [13]. PAPTP and PCARBTP have been shown to be effective in an orthotopic mouse model of malignant melanoma and pancreas carcinoma and in a genetic model of B-chronic lymphocytic leukemia without inducing apoptosis in healthy organs or cardiotoxicity and were well tolerated [13,21]. However, the effectivity of PAPTP and PCARBTP in a mouse model of multiple myeloma has not been shown so far and needs to be tested.

The cytotoxic effect of Kv1.3 inhibitors in tumor cells is mediated by inhibition of the depolarizing K+ influx through the mitochondrial Kv1.3 channel. This leads to an initial hyperpolarization of the inner mitochondrial membrane, which in turn leads to a reduction in the respiratory chain and ATP production, resulting in the production of mitochondrial ROS. This results in the release of cytochrome c, opening of the permeability transition pore and finally apoptosis [13,17,18]. The proposed mechanism, why tumor cells are more vulnerable for cell death than healthy cells is explained by differences in the basal, preexisting ROS levels in mitochondria. The proapoptotic effect of Kv1.3 inhibitors is increased when the mitochondrial redox status is altered such that basal ROS levels are higher [30]. The already higher basal ROS levels enable—in the sense of a synergistic effect with inhibition of the Kv1.3 channels—the crossing of a critical ROS threshold which is necessary to initiate apoptosis [13]. In line with these findings, we detected an early increase in mitochondrial ROS and breakdown of the membrane potential in mitochondria after treatment of human malignant melanoma cells with PAPTP and PCARBTP (Figure 3a,b). Further, we detected an activation of caspase 3/7 and cleavage of PARP (not shown) as hallmarks of apoptotic cell death after treatment of human melanoma cell lines with PAPTP and PCARBTP (Figure 3c).

To identify resistance mechanisms against PAPTP and PCARBTP, we aimed to generate resistant plasma cell clones. Therefore, we incubated human plasma cell lines with increasing concentrations of PAPTP and PCARBTP. For PCARBTP, we were able to establish L-363 cells that were resistant to PCAPBTP up to 3.5 µM (L-363-R3.5) and RPMI cells that were resistant up to 1.0 µM (RPMI-8226-R1.0). Comparing gene expression in parental and PCARBTP-resistant cell clones, we solely identified MDR-1 as upregulated > 2-fold log change in both resistant cell clones (Appendix A), while we did not find any > 2-fold log change of a downregulated gene in both resistant cell clones (Appendix A). Along with MDR-1, five proteins involved in stress response that clustered together with MDR-1 in string analysis (HSPA8, HSP90AA1, HSPH1, HSPA1B, AHSA1) were upregulated, at least in RMPI-8226 cells. Inhibiting MDR-1 with elacridar, a specific inhibitor of MDR-1 [31], almost completely reconstituted sensitivity of the resistant cell clones to PCARBTP (Figure 4a,b), confirming that upregulation of MDR-1 is the main and predominant resistance mechanism against PCARBTP-induced cell death. However, while in L-363-R3.5 cells elacridar completely reconstituted sensitivity to PCARBTP, in RPMI-8226-R1.0 cells 25% remained resistant to cell death despite inhibition of MDR-1. Therefore, additional, not further defined resistance mechanisms may also play a role.

Another resistance mechanism could be the regulation of mitochondrial expression of Kv1.3 itself. Leanza et al. [13] experimentally demonstrated that down-regulation of Kv1.3 by appropriate siRNA can significantly decrease the sensitivity of cells to Kv1.3 inhibitors. In the resistant cell lines generated in the present study, we did not observe a significant change in the expression of Kv1.3 at the RNA level. Furthermore, the induced resistance could be almost completely reverted by inhibition of MDR-1 indicating that modulation of Kv1.3 expression seems to play a rather minor role.

Different levels of Kv1.3 expression might potentially explain the observation that the EC50 of PAPTP-induced cell killing in our primary plasma cells ranged from <0.01 to 5.33 µmol/L. However, due to the limited amount of plasma cells in the bone marrow aspirates obtained, we were unable to quantify mitochondrial expression of Kv1.3 in the plasma cells and therefore cannot exclude this potential resistance mechanism.

With increasing lines of therapy, there may be increased expression of MDR-1 [32], in particular carfilzomib-resistant multiple myeloma cells show a strong up-regulation of MDR-1 [33,34]. We have shown that PCARBTP, and to a much lesser extent also PAPTP, are substrates for MDR-1. This could have the consequence that perhaps in later therapy lines the effectiveness of Kv1.3 inhibitors could decrease. To what extent this will really be clinically relevant is presently difficult to assess and remains speculative.

Of note, despite several attempts, we were not able to generate PAPTP-resistant cell clones. Cells died when reaching a dose corresponding the EC50 of the parental cells. Further, we found cross-resistance of PCARPTP-resistant cells to PAPTP. However, this was significantly less pronounced than resistance to PCARBTP. While the EC50 of PAPTP increased from 0.09 µM in the parental RPMI-8226 cells to 0.19 µM in the RPMI-8226-R1.0 cells and thus approximately doubled (Figure 1b and Figure 4c), the RPMI-8226-R1.0 cells showed a nine-fold increase in EC50 with PCARBTP compared to the parental RPMI-8226 cells (Figure 1c and Figure 4a). The EC50 of PAPTP in the parental L-363 cells was 0.29 µM and increased to 0.75 µM in the L-363-R3.5 cells (Figure 1b and Figure 4d), corresponding to a two- to three-fold increase. In contrast, the L-363-R3.5 cells showed more than four- to five-fold increase in EC50 with PCARBTP compared to the parental RPMI-8226 cells (Figure 1c and Figure 4b).

These data indicate that, although cross-resistance to PAPTP exists in PCARBTP resistant cells, it is much less pronounced. Furthermore, it should be emphasized that this resistance to PAPTP can be completely reversed by inhibition of MDR-1 with elacridar; thus, this resistance can be completely explained by upregulation of MDR-1. The lower expression of resistance to PAPTP compared with PCARBTP suggests that PAPTP may be a poorer substrate for MDR-1 than PCARBTP. This may also be the reason why we could not generate PAPTP-resistant cell lines by incubating the parental cells with PAPTP.

The BCL-2 family proteins are key regulators of the mitochondrial apoptotic pathway and consist of pro- and antiapoptotic proteins. Bcl-2 is an anti-apoptotic protein that retains Bax, a pro-apoptotic protein, and therefore prevents Bax from inducing apoptosis [35]. Our previous work has shown that Kv1.3 located to the inner mitochondrial membrane mediates Bax-induced apoptosis. Bax binds to Kv1.3 and functionally inhibits mitochondrial Kv1.3 [17,18]. Therefore, we speculated that co-administration of ABT-199 (Venetoclax), a small drug that binds Bcl-2 specifically [28] and therefore releases Bax, could possibly act synergistically to Kv1.3 inhibitors. The efficacy of ABT-199 has been shown to be dependent on the expression-profile of Bcl-2, Bcl-xL and Mcl-1, high expression of Bcl-2 and lower expression of Bcl-xl and Mcl-1 render cells susceptible to ABT-199 toxicity [36]. This pattern is predominantly present in multiple myeloma cells harboring t(11;14) and it has been shown in vitro but also in patients with multiple myeloma that tumor cells harboring t(11;14) are susceptible to venetoclax treatment [5,37]. However, venetoclax has very limited activity in myeloma patients without t(11;14), which is the case in around 4/5 of multiple myeloma patients. Of note, despite all tested multiple myeloma cell lines being negative for t(11;14), we found that co-administration of ABT-199 (Venetoclax) with Kv1.3 inhibitors has a strong synergistic effect. Therefore, combining venetoclax with Kv1.3 inhibitors could potentially augment venetoclax activity in patients without t(11;14).

It should further be noted that in the group of patients without t(11;14) there may also be subgroups, such as patients with hyperdiploidy or low bone disease subtypes, which show a Bcl-2 expression pattern that predicts a good response to venetoclax (http://doi.org/10.1182/blood.V128.22.5613.5613, accessed on 4 April 2022). However, this needs to be further studied by clinical data in the future.

It has been shown that treatment with bortezomib in vitro leads to up-regulation of NOXA, a pro-apoptotic factor that neutralizes Mcl-1 [36]. This may also lead to disinhibition of Bax [38]. Therefore, a synergistic effect of Kv1.3 inhibition and treatment with bortezomib or directly with Mcl-1 inhibitors is conceivable. However, this should be investigated in detailed future studies.

## 5. Conclusions

We identify mitochondrial Kv1.3 channels as druggable targets against multiple myeloma. The Kv1.3 inhibitors PAPTP and PCARBTP are efficient against both tested human multiple myeloma cell lines and against ex vivo-cultured, patient-derived myeloma cells, while healthy bone marrow cells are spared from toxicity. Cell death after treatment with PAPTP and PCARBTP occurs via the mitochondrial apoptotic pathway. We identify MDR-1 as the main potential resistance mechanism to the latter drug. Combination with ABT-199 has a synergistic effect, suggesting that mitochondrial Kv1.3 inhibitors could potentially be used as combination partner to venetoclax also in the treatment of t(11;14) negative multiple myeloma. However, effectivity of PAPTP and PCARBTP in a mouse model of multiple myeloma has not been shown so far and needs to be tested.

## Figures and Tables

**Figure 1 cancers-14-01955-f001:**
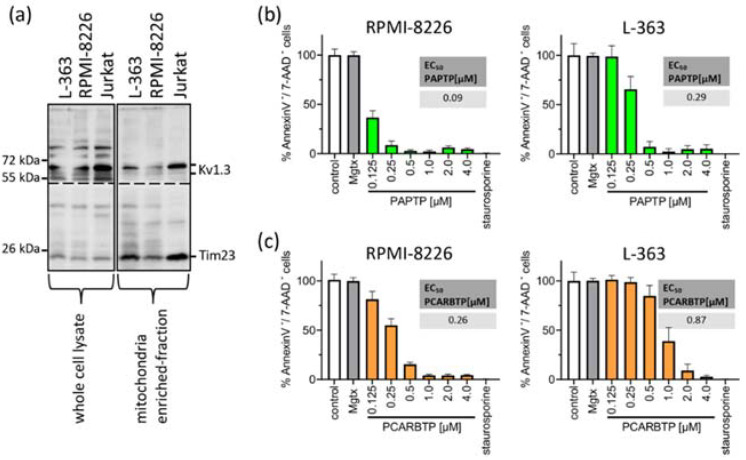
Mitochondrial Kv1.3 is expressed in the multiple myeloma cell lines L-363 and RPMI-8226. PAPTP and PCARBTP efficiently kill human multiple myeloma cells. (**a**) Whole cell lysates (left, 50 µg protein/lane) were lysed in RIPA-Buffer, mitochondria from L-363, RPMI-8226 and Jurkat cells (right, 30 µg protein/lane) were enriched as described; both were supplemented with 5× reducing SDS sample buffer, boiled and separated on 8.5% SDS-polyacrylamide gels and blotted on nitrocellulose membranes. Membranes were divided and, after blocking, membranes were incubated 1 h with anti-Kv1.3 or anti-Tim23 primary antibodies, respectively. After extensively washing, blots were incubated with secondary antibodies for 1 h, washed and developed with the Roti–Lumin system. Multiple myeloma cell lines RPMI-8226 and L-363 were treated with increasing concentrations PAPTP (**b**) or PCARBTP (**c**), solvent control (0.1% DMSO), 2 µM staurosporine as a positive control and 2 µM margatoxin as a membrane-impermeable Kv1.3 blocker. After 24 h, cells were stained with Annexin V-APC/7AAD and examined for cell death by flow cytometry. Results are reported as percentages with respect to untreated samples ± SD, (*n* = 3 independent experiments, each experiment in triplicate). Graph insert EC50: EC50 values of the indicated compounds in RPMI-8226 and L-363 were calculated by using GraphPad Prism version 9.3 for Windows (GraphPad Software, San Diego, CA, USA).

**Figure 2 cancers-14-01955-f002:**
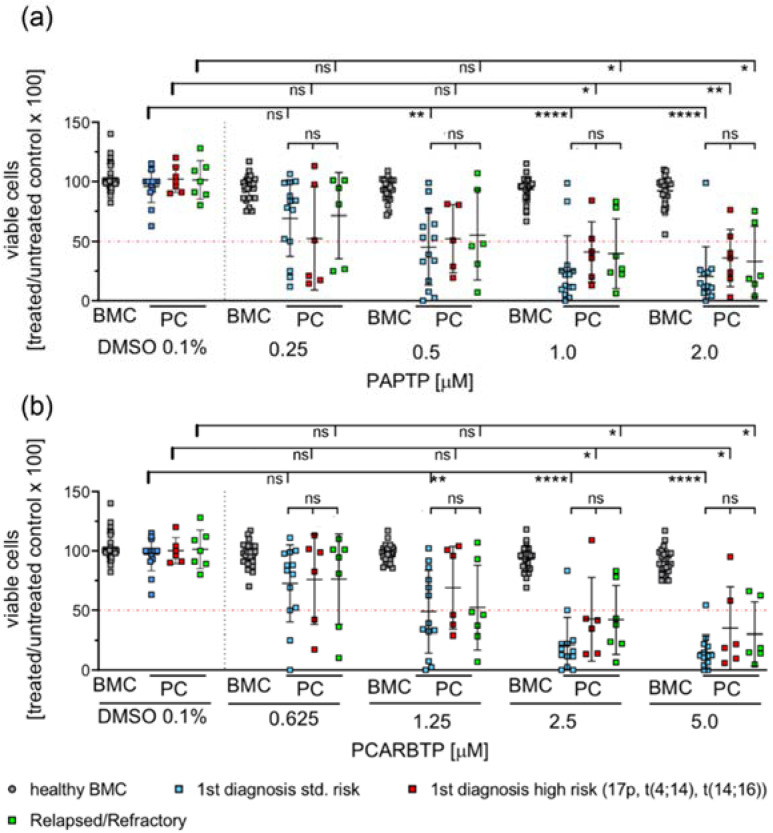
PAPTP and PCARBTP efficiently kill patient-derived myeloma cells ex vivo, while healthy bone marrow cells are spared from toxic effects. Isolated mononuclear bone marrow cells were cultured in IMDM 20% FCS at a concentration of 2 × 10^6^/mL and incubated with PAPTP, PCARBTP as indicated (**a**,**b**), respectively or 0.1% DMSO (solvent control). After 24 h cells were harvested and stained with Annexin V-APC and 7-AAD for cell death and simultaneously for CD19, CD38, CD138, CD45, CD269 and CD56 to properly detect myeloma cells. Samples were analyzed by flow cytometry, malignant plasma cells were defined as CD38^+^, CD138^+^, CD19^−^, CD269^+^ and CD56^+^. Shown is the mean ± SD of viable cells in % of untreated control, each dot representing the measure of one patient sample, *n* = 7–14, ns not significant, * *p* < 0.05, ** *p* < 0.01, **** *p* < 0.0001 Non-parametric ANOVA followed by Dunn’s multiple comparison and a Kruskal–Wallis test. ns: not significant.

**Figure 3 cancers-14-01955-f003:**
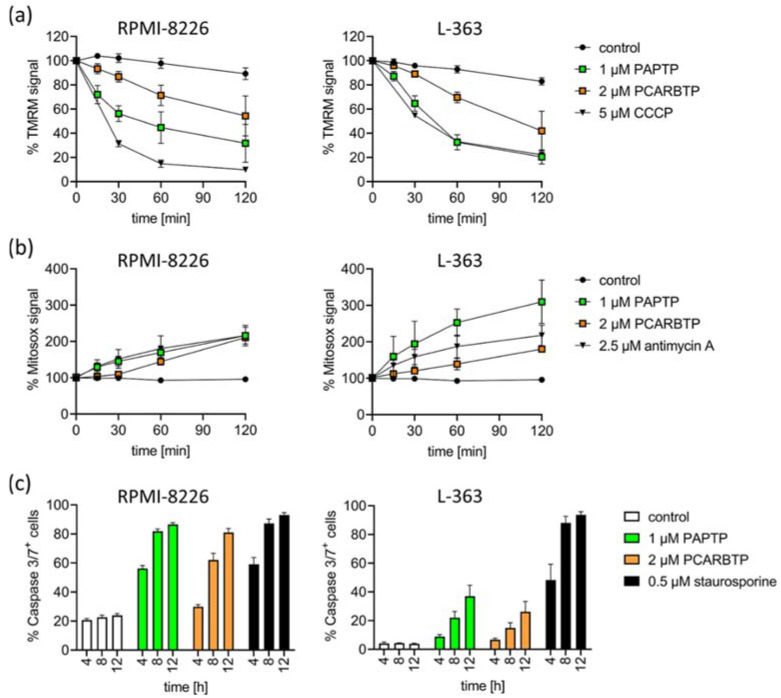
Inhibition of mitoKv1.3 with PAPTP and PCARBTP triggers apoptosis in human multiple myeloma cell lines L-363 and RPMI-8226. (**a**) Dissipation of the mitochondrial membrane potential was monitored following the TMRM signal over time. CCCP was used as the positive control. (**b**) Mitochondrial ROS production was monitored by the development of MitoSOX fluorescence over time. Antimycin A was used as the positive control. Values are reported as percentage MFI with respect to solvent control. (**c**) Caspase 3/7-activation was measured by flow cytometry 4, 8 and 12 h after treatment with PAPTP (1 µM), PCARPTP (2 µM) or staurosporine (0.5 µM) as a positive control. Shown is the mean ± SD of % caspase 3/7+ cells of 3 independent experiments, each experiment in triplicate.

**Figure 4 cancers-14-01955-f004:**
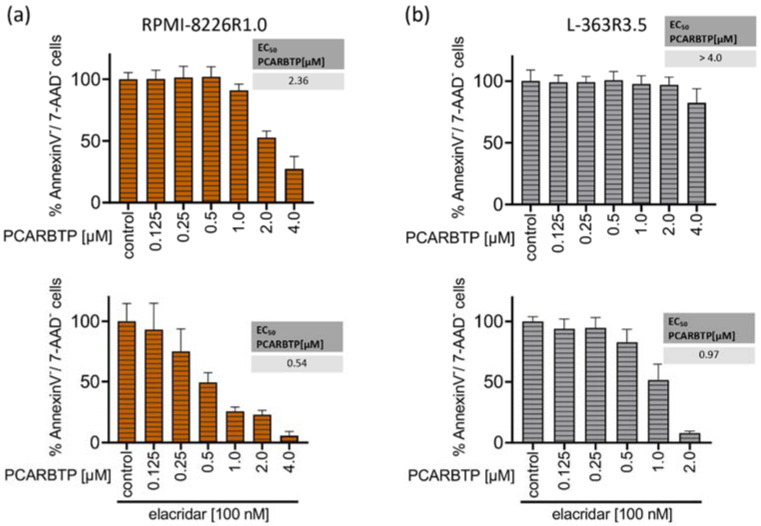
Resistance to PCARBTP is mediated by MDR-1, RPMI-8226-R1.0 and L-363-R3.5 are cross-resistant to PAPTP. (**a**,**b**) Parental L-363 and RPMI-8226 multiple myeloma cells and PCARBTP-resistant L-363-R3.5 and RPMI-8226-R1.0 cell lines were incubated with PCARBTP or 0.1% DMSO control in the presence or absence of elacridar as indicated. After 24 h, cell death was determined by flow cytometry with Annexin V-APC/7AAD-staining. Resistant RPMI-8226-R1.0 (**c**) and L-363-R3.5 cells (**d**) were treated with increasing concentrations PAPTP ± 100 nM elacridar for 24 h. After 24 h, cell death was determined by flow cytometry with Annexin V-APC/7AAD-staining. Values are normalized to solvent-treated control. Shown is the mean ± SD, (*n* = 3 independent experiments, each in duplicate).

**Figure 5 cancers-14-01955-f005:**
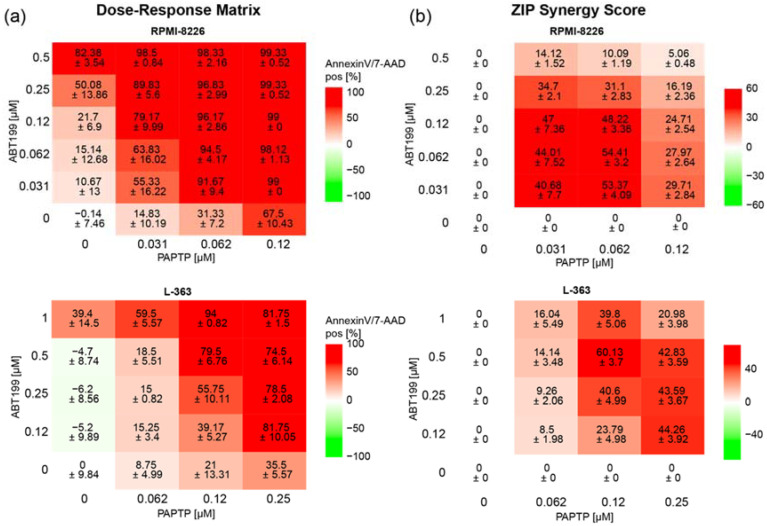
Combined treatment of multiple myeloma cell lines with PAPTP + ABT-199 improves killing compared to PAPTP alone. Cells were cultivated in the presence of different concentrations of PAPTP ± ABT-199 as indicated. After 24 h, cell death was analyzed by flow cytometry with Annexin V-APC/7AAD staining. (**a**) In the Dose–Response Matrix Annexin V/7AAD, positive cells are given in % normalized to solvent treated control. Shown is the mean ± SD, (*n* = 3 independent experiments, each in duplicates). (**b**) Synergies were calculated using the onlinetool https://synergyfinder.fimm.fi accessed on 24 February 2022 [26] and expressed as zero interaction potency-score [27].

## Data Availability

The data presented in this study are available in the article or in Appendix A. Raw RNA sequence data were uploaded to the NCBI Bioproject Database, accession number PRJNA809212, ID 809212.

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
