# Peer review of "Mitochondrial Kv1.3 Channels as Target for Treatment of Multiple Myeloma"

_cancers, 2022, doi:10.3390/cancers14081955_

Round 1

Reviewer 1 Report

This manuscript is a complete description of the potential interest of Kv1.3 channel  as a target for multiple myeloma.

  • Fig 1a: In RPMI -8226 Kv1.3 expression seems lower than in L-363 and than in the cell lysates. can you comment on that result?
  • Did you test mitochondrail Kv1.3 expression in plasma cells of patients? Is there a heterogeneity?
  • Kv1.3 inhibitor kill human plasma cell with a IC50from <0.01 to 5.33 micromol/L : Do you think there is a relation to Kv1.3 expresssion? Do you know if clinical outcome in patient with high IC 50 or non responsive was clinically different from others?
  • Patients who receive several lines of treatment may have increase expression of MDR. Does it change the potential interest of this target?
  • You studied bc-2 inhibitor as a potential synergic target. Do you think that proteasome inhibotor, with impact on MCL-1 should also be studied?

Author Response

Response to reviewer 1

This manuscript is a complete description of the potential interest of Kv1.3 channel  as a target for multiple myeloma.

  • Fig 1a: In RPMI -8226 Kv1.3 expression seems lower than in L-363 and than in the cell lysates. can you comment on that result?

Response: For mitochondrial Kv1.3 detection, 30 µg protein of the enriched mitochondrial fraction was used per cell line. The efficiency of mitochondrial enrichment (indicated by Tim23 bands) differed between cell lines. Normalization of the corresponding Kv1.3 bands to Tim23 results in approximately equal amounts of Kv1.3 in the mitochondria of the different cell lines. For the total cell lysates, more protein (50 µg) was applied per cell line. Please note, that Kv1.3 is not localized exclusively in mitochondria, but also in the plasma membrane and other cellular compartments. We provided additional information in the respective figure legend.

  • Did you test mitochondrial Kv1.3 expression in plasma cells of patients? Is there a heterogeneity?

Response: Unfortunately, due to limited cell numbers, we could not quantify Kv1.3 expression in patient plasma cells. To address this point, we would need to obtain and purify a much larger amount of cells from the patient or expand the cells in culture, which, however, could lead to artifacts. We added to the discussion following passage: ”Another resistance mechanism could be the regulation of mitochondrial expression of Kv1.3 itself. Leanza et al. [13] experimentally demonstrated that down-regulation of Kv1.3 by appropriate siRNA can significantly decrease the sensitivity of cells to Kv1.3 inhibitors. In the resistant cell lines generated in the present study, we did not observe a significant change in the expression of Kv1.3 at the RNA level. Furthermore, the induced resistance could be almost completely reverted by inhibition of MDR-1 indicating that modulation of Kv1.3 expression seems to play a rather minor role.

Different levels of Kv1.3 expression might potentially explain the observation that the EC50 of PAPTP-induced cell killing in our primary plasma cells ranged from <0.01 to 5.33 µmol/L. However, due to the limited amount of plasma cells in the bone marrow aspirates obtained, we were unable to quantify mitochondrial expression of Kv1.3 in the plasma cells and therefore cannot exclude this potential resistance mechanism.”

  • Kv1.3 inhibitor kill human plasma cell with a IC50 from <0.01 to 5.33 micromol/L : Do you think there is a relation to Kv1.3 expresssion?

Response: Please see the previous answer.

  • Do you know if clinical outcome in patient with high IC 50 or non responsive was clinically different from others?

Response: We added following sentences in Results: “We studied the 20 patients with initial diagnosis of multiple myeloma. For one patient we do not have sufficient data because hereceived further treatment elsewhere, another patient only received local radiotherapy, thereforetherapy responseafter chemotherapy and survival could notbe assessed here. For the remaining 18 patients, we were able to review the hematologic response to induction therapy and the best hematologic responses. The median follow-up time was 17 months, and at the time point of analysis17/18 patients were alive. All patients achieved at least a very good partial remission. We found no correlation between treatment response and in vitro sensitivity to Kv1.3 inhibition.”

  • Patients who receive several lines of treatment may have increase expression of MDR. Does it change the potential interest of this target?

Response: We agree with this point and added to the discussion following sentences: “With increasing lines of therapy, there may be increased expression of MDR-1 (Schwarzenbach 2002), in particular carfilzomib resistant multiple myeloma cells show a strong up-regulation of MDR-1 (Gutman D 2009) (Hawley TS 2013). We have shown that PCARBTP, and to a much lesser extent also PAPTP, are substrates for MDR-1. This could have the consequence that perhaps in later therapy lines the effectiveness of Kv1.3 inhibitors could decrease. To what extent this will really be clinically relevant is presently difficult to assess and remains speculative.”

  • You studied bc-2 inhibitor as a potential synergic target. Do you think that proteasome inhibitor, with impact on MCL-1 should also be studied?

Response: We agree with this point and added to the discussion following sentences: “It has been shown that treatment with bortezomib in vitroleads to up-regulation of NOXA, a pro-apoptotic factor that neutralizes MCL-1 (Punnoose et al. 2016). This may also lead to disinhibition of Bax (Germain M et al. 2008). Therefore, a synergistic effect of Kv1.3 inhibition and treatment with bortezomib or directly with Mcl-1 inhibitors is conceivable. However, this should be investigated in detailed future studies.

Reviewer 2 Report

Kadow et al. performed a study on mitochondrial Kv1.3 channels in MM. The topic sounds interesting and the manuscript is generally well written. However, some important issues should be addressed before further consideration.

  1. Theoretically, efficacy of venetoclax is dependent on Bcl-2 expression level regardless of t (11;14) status. High Bcl-2 is not limited to t(11;14). (Harrison et al. ASH 2019). This should also be discussed.
  2. The authors stated that healthy BM cells are not affected by Kv1.3 inhibition. Do you have any data on toxicities in other organ systems?  If the patients get severe non-hematologic toxicities, e.g. cardiotoxicity, neurotoxicity, hepatotoxicity, nephrotoxicity etc, it would be difficult to get an approval. This issue should be taken into account.
  3. Is there an established definition of resistance to Kv1.3 inhibitor (which threshold of PAPTP/PCARBTP concentration and percentage of surviving cells?). Please add references. This experiment has to be performed using an established method. Otherwise, the results are not plausible.
  4. Page 5, line 198: the authors should provide some more clinical characteristics of the patients included in this study. Pretreatments?
  5. Page 6-7: do cytogenetics and pretreatments have influence on features of mitochondria (which alteration? Which agent?)? If yes, please specify and add references.
  6. Figure 2: Do the parameters fulfil the requirement for ANOVA and post-hoc test? Normal distribution? Please specify
  7. The authors have already shown the data in Figure 2. Therefore, table 1 could be removed or put it into the supplementary file.
  8. ABCB1 (MDR-1) is not a specific resistance mechanism of Kv1.3 inhibitor (multi drug resistance - 1). This should be included into the discussion section.
  9. The authors observed variable expression of Kv1.3. Is there any relationship between Kv1.3 expression level and effect of Kv1.3 inhibition? Could low Kv1.3 also be a resistance mechanism?
  10. Do the authors have any data on synergistic effect of co-administration of Mcl-1 inhibitor and Kv1.3 inhibitor? This would be interesting for the readers. Please add results if possible.

Author Response

Kadow et al. performed a study on mitochondrial Kv1.3 channels in MM. The topic sounds interesting and the manuscript is generally well written. However, some important issues should be addressed before further consideration.

  1. Theoretically, efficacy of venetoclax is dependent on Bcl-2 expression level regardless of t (11;14) status. High Bcl-2 is not limited to t(11;14). (Harrison et al. ASH 2019). This should also be discussed.

Response: We added in the discussion: “It should further be noted that in the group of patients without t(11;14) there are also subgroups, such as patients with hyperdiploidy or low bone disease subtypes, which show a BCL-2 expression pattern that predicts a good response to venetoclax (Wu et al, ASH abstract 2016). However, this needs to be further studied by clinical data in the future.”

  1. The authors stated that healthy BM cells are not affected by Kv1.3 inhibition. Do you have any data on toxicities in other organ systems?  If the patients get severe non-hematologic toxicities, e.g. cardiotoxicity, neurotoxicity, hepatotoxicity, nephrotoxicity etc, it would be difficult to get an approval. This issue should be taken into account.

Response: We have modified the text as follows: “PAPTP and PCARBTP have been shown to be effective in an orthotopic mouse model of malignant melanoma and pancreas carcinoma and in a genetic model of B-chronic lymphocytic leukemia without inducing apoptosis in healthy organs or cardiotoxicity and were well tolerated [10, 18].”

  1. Is there an established definition of resistance to Kv1.3 inhibitor (which threshold of PAPTP/PCARBTP concentration and percentage of surviving cells?). Please add references. This experiment has to be performed using an established method. Otherwise, the results are not plausible.

Response: We provide more details in the Methods section and now define the criteria of resistant cells. We replaced the respective part from line 127: “Resistant RPMI-8226 and L-363 cells were generated by culture with increasing amounts of either PAPTP or PCARBTP according to a established protocol (Nitta A 1997). The starting dosage of PAPTP or PCARBTP was 1/10 of the EC50, medium was exchanged every other day. Every 1-2 weeks concentrations of PAPTP or PCARBTP were increased by 20%. Finally, cells were designated as resistant to a certain dosage when (i) the resultant cell line grew exponentially in the continuous presence of the respective concentrations of PCARBTP (1 µM in the case of RPMI-8226R1.0 and 3.5 µM in the case of L-363R3.5) and (ii) showed a long-term stability after removal of PCARBTP for several weeks (up to 90 days) and (iii) remained resistant after freeze/thaw cycles (McDermott M 2014)”.

Further, we removed the sentence in lines 312-319 (Results) because it is already stated in Methods: The initial dose of PAPTP or PCARBTP was 1/10 of the EC50, and the medium was changed every other day. Every 7 days, the concentrations of PAPTP or PCARBTP were increased by 20%.”

In addition we removed in lines 322-326 following part of the sentence because we felt it being confusing rather than giving further support: “…meaning that both cell lines could be treated with the around 4 fold IC50 dosage of the respective parental cells without showing cell death (3,85 fold parental IC50 for RPMI-8226-R1.0 and 4,02 fold parental IC50 for L-363-R3.5, please compare IC50-values in Figure 1 c and Figure 4 a, b) without relevant induction of cell death.

  1. Page 5, line 198: the authors should provide some more clinical characteristics of the patients included in this study. Pretreatments?

Response: We inserted the following part in Methods: “Samples from patients were collected between June 2019 and April 2021. Twenty patients were initial diagnoses of multiple myeloma and 7 patients had relapsed or progressed disease. The relapsed or progressive patients were previously treated with 1 to 4 lines of therapy, the age of the patients ranged from 37 to 79 years, with a median age of 62 years. 6 of the 7 pretreated patients had been previously induced with bortezomib-containing therapy and subsequently received high-dose therapy with autologous hematopoietic stem cell transplantation and maintenance therapy with lenalidomide as part of primary treatment. 3 of the 7 patients received combination therapies with daratumumab and imides or proteasome inhibitors in the primary or relapse therapy.”

  1. Page 6-7: do cytogenetics and pretreatments have influence on features of mitochondria (which alteration? Which agent?)? If yes, please specify and add references.

Response: Previous published studies have shown that resistance to proteasome-inhibitors has been connected to changes in mitochondrial metabolism. To our knowledge, changes in mitochondria in dependence of the presence or absence of high-risk cytogenetics are so far not described. We added the following section to the introduction:

“Mitochondria play essential roles in numerous cellular functions and in homeostasis, such as ATP production, the regulation and initiation of apoptosis, and in several metabolic adaptations. Multiple myeloma cells generally have upregulated protein biosynthesis as part of immunoglobulin synthesis and myeloma cells are therefore dependent on degrading misfolded, dysfunctional synthesized proteins as part of the Unfolded Protein Response (Walter P 2011, Reimold AM 2001). Treatment with proteasome inhibitors, such as bortezomib or carfilzomib, results in decreased degradation of these misfolded, dysfunctional proteins, leading to proteotoxic stress, and subsequently to activation of the Unfolded Protein Response, leading to apoptosis (Walter P 2011). It has been shown in vitrothat cell lines resistant to proteasome inhibitors have a distinctly upregulated chaperone machinery to process the misfolded proteins, for which adaptations in energy and mitochondrial metabolism are also necessary and have been described (Soriano GP 2016). In addition to these changes, upregulation of the multi-drug resistance protein (MDR-1) ABCB1 was detected, particularly in cell lines where resistance to carfilzomib was generated, and carfilzomib resistance could be abrogated by inhibition of MDR-1 (Soriano GP 2016).”

  1. Figure 2: Do the parameters fulfil the requirement for ANOVA and post-hoc test? Normal distribution? Please specify

Response: We re-evaluated the data and in fact some data partially deviated from a normal distribution. We therefore modified the statistical analysis and used non-parametric ANOVA followed by Dunn's multiple comparison and Kruskal-Wallis test. We changed this in Figure Legend 2 accordingly.

  1. The authors have already shown the data in Figure 2. Therefore, table 1 could be removed or put it into the supplementary file.

Response: We moved Table 1 from the main manuscript to the supplements.

  1. ABCB1 (MDR-1) is not a specific resistance mechanism of Kv1.3 inhibitor (multi drug resistance - 1). This should be included into the discussion section.

Response: We agree with this point and added to the discussion following sentences: “With increasing lines of therapy, there may be increased expression of MDR-1 (Schwarzenbach 2002), in particular carfilzomib resistant multiple myeloma cells show a strong up-regulation of MDR-1 (Gutman D 2009) (Hawley TS 2013). We have shown that PCARBTP, and to a much lesser extent also PAPTP, are substrates for MDR-1. This could have the consequence that perhaps in later therapy lines the effectiveness of Kv1.3 inhibitors could decrease. To what extent this will really be clinically relevant is presently difficult to assess and remains speculative.”

  1. The authors observed variable expression of Kv1.3. Is there any relationship between Kv1.3 expression level and effect of Kv1.3 inhibition? Could low Kv1.3 also be a resistance mechanism? 

Response: Unfortunately, due to limited cell numbers, we could not quantify Kv1.3 expression in patient plasma cells. To address this point, we would need to obtain and purify a much larger amount of cells from the patient or expand the cells in culture, which, however, could lead to artifacts. We added to the discussion following passage: ”Another resistance mechanism could be the regulation of mitochondrial expression of Kv1.3 itself. Leanza et al. [13] experimentally demonstrated that down-regulation of Kv1.3 by appropriate siRNA can significantly decrease the sensitivity of cells to Kv1.3 inhibitors. In the resistant cell lines generated in the present study, we did not observe a significant change in the expression of Kv1.3 at the RNA level. Furthermore, the induced resistance could be almost completely reverted by inhibition of MDR-1 indicating that modulation of Kv1.3 expression seems to play a rather minor role.

Round 2

Reviewer 2 Report

I have no further comments